# Safety Evaluation by Phenotypic and Genomic Characterization of Four *Lactobacilli* Strains with Probiotic Properties

**DOI:** 10.3390/microorganisms10112218

**Published:** 2022-11-09

**Authors:** Ye-Rim Lee, Won Yeong Bang, Kwang-Rim Baek, Geun-Hyung Kim, Min-Ji Kang, Jungwoo Yang, Seung-Oh Seo

**Affiliations:** 1Department of Food Science and Nutrition, The Catholic University of Korea, Bucheon 14662, Korea; 2Ildong Bioscience, Pyeongtaek-si 17957, Korea

**Keywords:** *Lactobacilli*, *Lactobacillus fermentum*, *Lactobacillus gasseri*, *Lactobacillus helveticus*, *Lactobacillus salivarius*, probiotics, safety evaluation

## Abstract

Probiotic *Lactobacillus* species are known to exert health benefits in hosts when administered in adequate quantities. A systematic safety assessment of the strains must be performed before the *Lactobacillus* strains can be designated as probiotics for human consumption. In this study, we selected *Lactobacillus fermentum* IDCC 3901, *L. gasseri* IDCC 3101, *L. helveticus* IDCC 3801, and *L. salivarius* IDCC 3551 as representative *Lactobacilli* probiotic strains and investigated their probiotic properties and potential risks through phenotypic and genomic characterization. Various assays including antimicrobial resistance, biogenic amine production, L-/D-lactate production, acute oral toxicity, and antipathogenic effect were performed to evaluate the safety of the four *Lactobacillus* strains. Genomic analysis using whole genome sequencing was performed to investigate virulence and antibiotic resistance genes in the genomes of the selected probiotic strains. The phenotypes of the strains such as enzymatic activity and carbohydrate utilization were also investigated. As a result, antibiotic resistances of the four *Lactobacillus* species were detected; however, neither antibiotic resistance-related genes nor virulence genes were found by genomic analysis. Moreover, the four *Lactobacillus* species did not exhibit hemolytic activity or β-glucuronidase activity. The biogenic amine production and oral acute toxicity were not shown in the four *Lactobacillus* species, whereas they produced D-lactate with minor ratio. The four *Lactobacillus* species exhibited antipathogenic effect to five pathogenic microorganisms. This study provides a way to assess the potential risks of four different *Lactobacillus* species and validates the safety of all four strains as probiotics for human consumption.

## 1. Introduction

Probiotics are live microorganisms that confer health benefits to the host when administered in adequate amounts [1]. Lactic acid bacteria (LAB), including *Lactobacilli*, are well-known probiotics. *Lactobacilli* are Gram-positive, rod-shaped bacteria. *Lactobacilli* are traditionally used for fermentation of dairy, vegetables, and beverages [2]. In dairy industry, *L. helveticus*, *L. delbruekii* subsp. *bulgaricus*, *L. paracasei*, and *L. rhamnosus* are important in cheese fermentation [3]. *L. delbruekii* subsp. *bulgaricus* is also a traditional yogurt bacterium [4]. *L. plantarum and L. brevis* are usually isolated from fermented vegetables [5]. Moreover, wine fermentation includes some *Lactobacilli* such as *L. plantarum*, *L. brevis*, *L. paracasei* [6]. *Lactobacilli* are important members of the human gut microbiota, and some strains are known for their varied health benefits [7]. *Lactobacillus* species are extensively studied for their intestinal regulation by immunomodulatory mechanism [8]. Several representative strains of *Lactobacilli* with probiotic potential have been consumed as dietary supplements including *L. rhamnosus*, *L. helveticus*, *L. fermentum*, *L. gasseri*, *L. bulgaricus*, *L. acidophilus*, *L. casei*, and *L. reuteri* [9].

Various biological activities of probiotic nature have been reported in *Lactobacilli* species, and the demand for such probiotic lactobacilli species for intake in promoting human health has been present for decades. Therefore, oral intake of probiotic *Lactobacilli* should be proven safe for humans by a systematic safety assessment. Genome analysis can be used for performing safety evaluation because it can provide genetic information about probiotic strains, such as the presence of virulence genes and antibiotic resistance genes in their genome.

Previously, we showed that four probiotics strains, *L. fermentum* IDCC 3901, *L. gasseri* IDCC 3101, *L. helveticus* IDCC 3801, and *L. salivarius* IDCC 3551 have 76-87% acid tolerance, 96–97% bile tolerance, 42–53% adhesion to Caco-2 cells, and 70–72% competitive exclusion against pathogenic bacteria [10]. Furthermore, we applied quadruple coating to these strains to improve these probiotics properties in the manufacturing process [10,11]. Fermented milk by *L. helveticus* IDCC 3801 alleviated memory deficit by reducing beta-amyloid in a rat model [12]. In this study, four *Lactobacillus* strains, *L. fermentum* IDCC 3901, *L. gasseri* IDCC 3101, *L. helveticus* IDCC 3801, and *L. salivarius* IDCC 3551, were selected and evaluated. This study provides a way to assess the potential risks of four *Lactobacillus* species using various phenotypic and genomic analyses, including whole genome sequence analysis, minimum inhibitory concentration (MIC) test, β-hemolytic activity assay, enzymatic activity assay, carbohydrate utilization test, biogenic amine production analysis, L-/D-lactate production analysis, and oral acute toxicity tests. Finally, the antipathogenic activities of the strains were investigated in terms of preventing or relieving chronic inflammation in the gastrointestinal tract. Thus, the systematic analysis performed in this study validates the use of these four specific *Lactobacillus* strains as safe probiotic strains.

## 2. Materials and Methods

### 2.1. Bacterial Strains and Growth Conditions

*L. fermentum* IDCC 3901, *L. gasseri* IDCC 3101, *L. helveticus* IDCC 3801, and *L. salivarius* IDCC 3551 were stored at −80 °C by Ildong Bioscience (Pyeongtaek-si, Gyeonggi-do, Korea). All strains were grown in De Man, Rogosa, and Sharpe (MRS) medium (BD Difco, Franklin Lakes, NJ, USA) at 37 °C. All four *Lactobacillus* strains were used for probiotic manufacturing at Ildong Bioscience (Table 1).

For the antipathogenic experiment, the five pathogenic microorganisms including *Staphylococcus aureus* ATCC 25923, *Enterococcus faecalis* ATCC 29212, *Streptococcus pneumonia* ATCC 49619, *Bacillus cereus* ATCC 14579, and *Salmonella* Typhimurium ATCC 13311 were purchased from the American Type Culture Collection (ATCC). The culture conditions of each pathogenic microorganism are listed in Table 2. 

### 2.2. Genomic Analysis

Genomic DNA of the *Lactobacillus* strains *L. fermentum* IDCC 3901, *L. gasseri* IDCC 3101, and *L. helveticus* IDCC 3801 was extracted using a Maxwell 16 LEV Blood DNA Kit and a Maxwell 16 Buccal Swab LEV DNA Purification Kit (Promega Co., Madison, WI, USA) according to the manufacturer’s instructions. Genomic DNA was extracted from the *L. salivarius* IDCC 3551 using the Wizard Genomic DNA Purification Kit (Promega Co., USA) according to the manufacturer’s instructions. Genome sequencing was performed by Macrogen Inc. (Korea) using a PacBio RS II instrument (Pacific Biosciences of California Inc., Menlo Park, CA, USA) on an Illumina platform (Illumina Inc., San Diego, CA, USA). The average nucleotide identity (ANI) value was calculated using an ANI calculator (Kostas Lab).

### 2.3. Identification of Antibiotic Resistance, Virulence Genes, and Mobile Elements

To identify putative antibiotic resistance genes, the assembled sequences were compared to the reference sequences in the ResFinder database (https://cge.cbs.dtu.dk/services/ResFinder/) using ResFinder 3.2 software (accessed on 1 July 2021) [13]. The search parameters for the analysis were sequence identity of >80% and coverage of >60%. To analyze virulence genes, VFDB (http://www.mgc.ac.cn/VFs/, accessed on 1 July 2021) was used with the BLASTn algorithm [14]. The thresholds for identification were as follows: identity > 70%, coverage > 70%, and E-value < 1E-5. In addition, mobile elements were searched using the BLASTP algorithm (for transposases and plasmids) and the PHASTER web-based program (for prophage regions) [15].

### 2.4. Determination of Minimum Inhibitory Concentrations (MICs)

The four *Lactobacillus* strains were assessed for susceptibility to ampicillin, vancomycin, gentamicin, kanamycin, streptomycin, erythromycin, clindamycin, tetracycline, and chloramphenicol (Sigma-Aldrich, St. Louis, MO, USA). Cultures of each strain and antibiotics were mixed in a 96-well microplate and anaerobically incubated at 37 °C for 18–20 h. The optical density was measured using a microplate reader (BioTek, Winooski, VT, USA).

### 2.5. β-Hemolytic and Enzymatic Activities

Each *Lactobacillus* species was grown overnight at 37 °C in MRS medium, and the culture was streaked on sheep blood agar plates (BD Difco). The plates were incubated overnight at 37 °C. A clear zone around the colony demonstrated hemolytic activity. For all samples, *Staphylococcus aureus* subsp. *aureus* ATCC 25923 was used as a positive control.

The enzyme activities of all four strains were determined using an API ZYM kit (BIOMÉRIUX, Marcy-l’Étoile, France) with 19 different substrates. Briefly, the four *Lactobacillus* strains were cultured overnight at 37 °C in MRS medium. The cultures of each strain were harvested by centrifugation at 6000 rpm for 8 min at 4 °C and each cell pellet was resuspended in a sterile saline solution to adjust the final cell concentration to 1 × 10^9^ CFU/mL. The resuspended cells were inoculated into a well-type plate provided by the manufacturer, and the plate was incubated at 37 °C for 4 h. ZYM A and ZYM B solutions were then added to wells, and the color changes were observed after 5 min at room temperature.

### 2.6. Carbohydrate Utilization

The carbohydrate utilization ability was determined using an API 50 CHL/CHB Kit (BIOMÉRIUX, Marcy-l’Étoile, France) and 49 different carbohydrates. The cultures of four *Lactobacillus* strains were harvested by centrifugation at 6000 rpm for 8 min at 4 °C and each cell pellet (6 × 10^8^ CFU/mL) was resuspended in API 50 CHL medium. The resuspended cells were inoculated into a well-type plate provided by manufacturer, and the plate was incubated at 37 °C for 48 h. The color changes were then observed. 

### 2.7. Biogenic Amine (BA) Production

The supernatant from the overnight cultures of each *Lactobacillus* species was obtained by centrifugation at 6000 rpm and filtrated through a 0.22 µm pore size membrane. An aliquot of each supernatant (0.5 mL) was mixed with the same aliquot of 0.1 M HCl and filtered through a 0.45 µm membrane for extraction of BAs. For the derivatization, 1 mL of the extracted BAs was incubated at 70 °C for 10 min, which was followed by addition of 200 µL of saturated NaHCO_3_, 20 µL of 2 M NaOH, and 0.5 mL of dansyl chloride (10 mg/mL acetone). The derivatized BAs were mixed with 200 µL of L-proline (100 mg/mL H_2_O) and incubated in darkness at room temperature for 15 min. Acetonitrile (HPLC grade; Sigma-Aldrich, St. Louis, MO) was added to obtain a final volume of 5 mL. High-performance chromatography was performed to separate and quantify the BAs using an HPLC instrument (Agilent 1260, Agilent Technologies, CA, USA) equipped with a C18 column (YMC-Triart, 4.6 × 250 mm, YMC, Kyoto, Japan) and a UV detector (G7115A, Agilent Technologies, CA). Acetonitrile solution (acetonitrile: H_2_O = 67:33, *v*/*v*) was used as the mobile phase at a constant flow rate of 0.8 mL/min. Calibration of each BA, including tyramine, histamine, putrescine, 2-phenethylamine, cadaverine, and tryptamine, was used to quantify BAs (Sigma-Aldrich, St. Louis, MO, USA).

### 2.8. D-/L-Lactate Production

L- and D-lactate levels were measured using an assay kit (Megazyme, Bray, Ireland) according to the manufacturer’s protocol. The supernatant (0.1 mL) from the culture was mixed with 1.5 mL of H_2_O, 0.5 mL of buffer solution (pH 10.0), 0.1 mL of NAD^+^ solution, and 0.02 mL of glutamate-pyruvate transaminase (GPT) and incubated at room temperature for 3 min. Subsequently, 0.02 mL of lactate dehydrogenase (LDH; 2000 U/mL) was added to the reaction mixture. The absorbance of D-lactate was measured at 340 nm until the reaction stopped. The concentrations of L- and D-lactate were then calculated using equations from the manufacturer’s protocol.

### 2.9. Acute Oral Toxicity

Acute oral toxicity tests were performed at the Korea Testing and Research Institute (Hwasun-gun, Jeollanam-do, Korea). The rats used in this study were bred under the following environmental conditions: 21.1–22.3 °C, 40.5–58.0% relative humidity, 12 h-light/dark cycle, 150–300 Lux of illumination, 270 × 500 × 200 mm (W× D × H) cage size, and less than three rats per cage. The rats were allowed free access to food (Rodent Diet 20 5053; Labdiet, St. Louis, MO, USA) and water. Four groups of female rats were divided by age (9–10 weeks) and administered doses (mg/kg B.W.) of *Lactobacilli* powder solution in sterile distilled water. Clinical signs, changes in body weight, and necropsy findings were investigated during the test period of 14 days.

### 2.10. Antipathogenic Effect

The four *Lactobacillus* strains were grown overnight at 37 °C in MRS medium. The cell-free supernatant was obtained using centrifugation at 6000 rpm for 8 min at 4 °C and filtered through a 0.22 μm pore size membrane. Five pathogenic microorganisms, including *Staphylococcus aureus* ATCC 25923, *Enterococcus faecalis* ATCC 29212, *Streptococcus pneumonia* ATCC 49619, *Bacillus cereus* ATCC 14579, and *Salmonella* Typhimurium ATCC 13311, were adjusted to an initial cell density of 1.5 x 10^8^ CFU/mL using the McFarland standard (BioMerieux, Marcy-I’Etoile, France). The cell-free supernatant of each strain was then mixed with each pathogen. The mixture was incubated in a 96-well microplate under the optimal culture conditions of each pathogen. The optical density was measured using a microplate reader at time 0 and 24 h. The antimicrobial effect of the four *Lactobacillus* strains was determined by complete inhibition of the cell growth of pathogens. 

## 3. Results

### 3.1. Whole Genome Sequence Analysis

Whole genome sequencing was performed for all four of the selected *Lactobacillus* strains, *L. fermentum* IDCC 3901, *L. gasseri* IDCC 3101, *L. helveticus* IDCC 3801, and *L. salivarius* IDCC 3551. Based on these results, the four *Lactobacillus* strains were confirmed as *Limosilactobacillus fermentum* (previously classified as *Lactobacillus fermentum*), *L. gasseri*, *L. helveticus*, and *L. salivarius*, respectively. Information regarding the size of the genome, GC content, and CDSs is listed in Table 3. The genome sequences of the four strains were analyzed to identify antibiotic resistance and virulence genes. None of the four strains had genes related to general antibiotic resistance, such as aminoglycosides, beta-lactams, MLS-macrolide, lincosamide, phenicol, and tetracycline (Table 4). According to the BLASTn algorithm and VFDB, no virulence genes were found in the genomes of the four *Lactobacillus* strains. Thus, genome sequencing results indicated the safety of *L. fermentum* IDCC 3901, *L. gasseri* IDCC 3101, *L. helveticus* IDCC 3801, and *L. salivarius* IDCC 3551 for human consumption.

### 3.2. Determination of MICs

MIC tests were performed for the four *Lactobacillus* species with ampicillin, vancomycin, gentamicin, kanamycin, streptomycin, erythromycin, clindamycin, tetracycline, and chloramphenicol. *L. fermentum* IDCC 3901, *L. gasseri* IDCC 3101, and *L. salivarius* IDCC 3551 were resistant to gentamicin, kanamycin, and streptomycin. However, *L. helveticus* IDCC 3801 was resistant only to gentamicin and kanamycin. Apart from these results, the four species were also susceptible to other antibiotics; according to genome sequence analysis, they exhibit intrinsic antibiotic resistance to certain antibiotics. Intrinsic antibiotic resistance against aminoglycosides, such as gentamicin, kanamycin, and streptomycin, has been found in many *Lactobacillus* species [16]. Despite the presence of antibiotic resistance as revealed through the MIC analysis, no gene related to resistance against these antibiotics was detected in the genome sequences of the four *Lactobacillus* strains.

### 3.3. β-Hemolytic and Enzyme Activity

Ideally, hemolytic activity must be absent in probiotic strains. Hemolytic activity is the ability to lyse red blood cells, resulting in the destruction of hemoglobin. Based on the β-hemolytic activity test, the four *Lactobacillus* species did not exhibit hemolytic activity. *L. fermentum* IDCC 3901, *L. gasseri* IDCC 3101, *L. helveticus* IDCC 3801, and *L. salivarius* IDCC 3551 did not show clear zones on sheep blood agar, whereas the positive control *S. aureus* subsp. *aureus* ATCC 25923 showed a clear zone (Figure 1).

The enzymatic activity assay included 19 enzymes involved in carbohydrate, lipid, and vitamin metabolism. The results of enzymatic activity are presented in Table 5. *L. salivarius* IDCC 3551 did not display esterase activity, whereas the other strains did. *L. fermentum* IDCC 3901 and *L. salivarius* IDCC 3551 were positive for valine arylamidase activity. *L. helveticus* IDCC 3801 and *L. salivarius* IDCC 3551 exhibited activity for cystine arylamidase. Only *L. fermentum* IDCC 3901 had naphthol-AS-BI-phosphohydrolase activity, whereas the other strains did not. *L. helveticus* IDCC 3801 did not exhibit ɑ-galactosidase activity, whereas the other strains did. Only *L. fermentum* IDCC 3901 showed ɑ-glucosidase activity, whereas the other strains did not. *L. gasseri* IDCC 3101 and *L. helveticus* IDCC 3801 exhibited activity for β-glucosidase. Only *L. gasseri* IDCC 3101 exhibited N-acetyl-β-glucosaminidase activity. β-Glucuronidase may be related to colon cancer because it produces carcinogenic compounds [17]. However, the four *lactobacilli* did not present β-glucuronidase activity, so they are free of the safety concerns of β-glucuronidase.

### 3.4. Carbohydrate Utilization

Lactic acid bacteria have different carbohydrate metabolisms. Thus, investigation of carbohydrate utilization in different probiotic strains can provide important phenotypic characterization. The API 50 CH test results for the four *Lactobacillus* strains are listed in Table 6. *L. fermentum* IDCC 3901 metabolized L-arabinose, ribose, gluconate, and 2-keto-gluconate, whereas the other species did not. *L. gasseri* IDCC 3101 metabolized d-turanose and d-tagatose. *L. salivarius* IDCC 3551 metabolizes mannitol, sorbitol, and l-arabitol. All four *Lactobacillus* strains could metabolize galactose, d-glucose, d-fructose, esculine, maltose, lactose, and sucrose.

### 3.5. Production of Biogenic Amine and Lactate

The four selected *Lactobacillus* strains were also evaluated for biogenic amine production because biogenic amines are potential health risks [18]. According to HPLC analysis, none of the four *Lactobacillus* species produced biogenic amines, including tyramine, histamine, putrescine, 2-phenethylamine, cadaverine, and tryptamine (Appendix A). 

Some lactic acid bacteria produce lactate with two isomers, L-lactate and D-lactate, depending on environmental conditions [19]. D-lactate cannot be metabolized in the human body; therefore, D-lactate can accumulate, resulting in acidosis [20]. However, recent studies have demonstrated that the accumulation of D-lactate occurs only in cases of impaired D-lactate metabolism [13]. The four *Lactobacillus* strains selected in this study produced both L- and D-lactate isomers at varying ratios (Appendix A). However, the predominant production of L-lactate compared with that of D-lactate was noticed in all four *Lactobacillus* strains, indicating that they were less problematic for consumption.

### 3.6. Acute Oral Toxicity

Acute oral toxicity was investigated using a single-dose acute oral toxicity test. No deaths occurred, and no clinical signs were observed in the rats during the study. There were no significant body weight changes in rats administered *Lactobacillus* species (Table 7). Furthermore, no abnormal necropsy findings were observed. These results indicate that the four *Lactobacillus* species did not negatively affect the health of the rats.

### 3.7. Antipathogenic Effects of the Four Lactobacillus Strains

Antimicrobial activity is an important criterion for the selection of microorganisms to be used as probiotics [21]. Here, the four *Lactobacillus* strains completely inhibited bacterial growth of intestinal and pneumonia pathogens, including *Staphylococcus aureus* ATCC 25923, *Enterococcus faecalis* ATCC 29212, *Streptococcus pneumonia* ATCC 49619, *Bacillus cereus* ATCC 14579, and *Salmonella* Typhimurium ATCC 13311 (Figure 2). Interestingly, the pH of the cell-free supernatant of *L. fermentum* IDCC 3901, *L. gasseri* IDCC 3101, *L. helveticus* IDCC 3801, and *L. salivarius* IDCC 3551 were 4.85 ± 0.95, 3.82 ± 0.04, 3.67 ± 0.03, and 3.90 ± 0.10, respectively. Considering that pKa value of acetic acid is 4.75, acetic acids produced by these strains would be antipathogenic substances [22]. Lactic acids produced by *L. fermentum* IDCC 3901, *L. gasseri* IDCC 3101, *L. helveticus* IDCC 3801, and *L. salivarius* IDCC 3551 were measured at 831.36 mg/L ± 26.42, 1520.70 mg/L ± 6.53, 79.67 mg/L ± 20.16, and 1585.03 mg/L ± 800.69, respectively. In addition, the *Lactobacillus* genus is known to produce bacteriocin, small antibacterial peptide [23]. Pathogens such as *S. aureus*, *E. faecalis*, *B. cereus*, and *S.* Typhimurium are known to cause gastrointestinal disease by destruction of the epithelial structures. Among them, *Salmonella* species are known as main food poisoning bacteria, causing mucosal ulceration [24]. Furthermore, *S. pneumonia* is a bacterium that affects the respiratory system, causing pneumonia, bronchitis, and otitis media [25]. Thus, these four strains would be suggested as health functional food in terms of preventing or relieving chronic inflammation in the gastrointestinal tract.

## 4. Discussion

According to the result of the genomic analysis in our study, genes related to antibiotic resistance or virulence were not found in the four *Lactobacillus* species. However, the strains were revealed to be resistant to some antibiotics by the MIC tests. The antibiotic resistance of some *Lactobacillus* species was studied from other several studies [26,27], and that of our four *Lactobacillus* species was comparable to these results. Some gram-positive bacteria have hemolytic activity for iron uptake and host defense [28]. β-Hemolytic activity, a complete lysis of hemoglobin, was not detected from any of the four *Lactobacillus* species. Bacterial β-glucuronidase might be related to colorectal cancer [29]; therefore, the β-glucuronidase activity should be tested for probiotic strains for human consumption. Our results showed that none of the four *Lactobacillus* strains had β-glucuronidase activity. Biogenic amines are tentatively harmful elements and have a recommended level in foods according to the Food and Drug Administration (FDA) [30]. Potential toxic compounds, iogenic amines are produced by some bacteria, which is why the biogenic amine production test was conducted for probiotic strains. In our study, none of the four *Lactobacillus* species produced biogenic amines. D-lactate has been considered to be a factor of acidosis in the human body because of its accumulation. However, D-lactate acidosis may not be a concern to healthy humans [13]. Acute oral toxicity is necessary to prove the safety of probiotic strains for human consumption. Due to the similarity of the physiology of rats and humans, oral toxicity tests using rats can provide evidence for the safety of human consumption [31]. In our oral toxicity test, none of the *Lactobacillus* strains occurred adverse events in the rats. The antipathogenic effect of the four *Lactobacillus* strains was investigated with *S. aureus*, *E. faecalis*, *S. pneumonia*, *B. cereus*, and *S.* Typhimurium which can cause gastrointestinal disease. In our study, our four *Lactobacillus* species inhibited the growth of the pathogenic bacteria. These results demonstrate that *L. fermentum* IDCC 3901, *L. gasseri* IDCC 3101, *L. helveticus* IDCC 3801, and *L. salivarius* IDCC 3551 can be considered safe as probiotic strains. For preclinical studies of these probiotic strains with a medical purpose, more experiments may be required to identify these strains’ acute and subacute toxicity, immunotoxicity, embrotoxicity, etc.

## 5. Conclusions

This study provides a safety assessment of four different *Lactobacillus* species, *L. fermentum*, *L. gasseri*, *L. helveticus*, and *L. salivarius*, through phenotypic and genomic analyses, including whole genome sequencing, MIC test, β-hemolytic and enzyme activity test, carbohydrate utilization test, analysis of biogenic amine and L-/D-lactate production, and acute oral toxicity test in rats. As a result, antibiotic resistances of the four *Lactobacillus* species were detected; however, neither antibiotic resistance-related genes nor virulence genes were found by genomic analysis. The selected four *Lactobacillus* species did not exhibit hemolytic activity or β-glucuronidase activity. Neither biogenic amine production nor oral acute toxicity was found in any strains. Moreover, the four *Lactobacillus* strains exhibited an antipathogenic effect against five pathogenic microorganisms. Through this systematic safety evaluation, we concluded that the four *Lactobacillus* strains selected and studied in this study may be utilized as a probiotic food supplement. Further in-depth functional study is required to elucidate the health benefits of these strains.

## Figures and Tables

**Figure 1 microorganisms-10-02218-f001:**
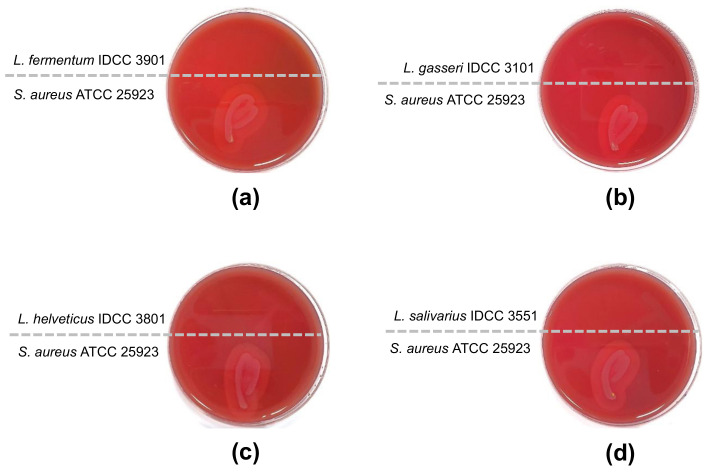
Test for β-hemolytic activity. *S. aureus* subsp. *aureus* ATCC 25923 was used for a positive control and showed β-hemolytic activity. (**a**) *L. fermentum* IDCC 3901; (**b**) *L. gasseri* IDCC 3101; (**c**) *L. helveticus* IDCC 3801; (**d**) *L. salivarius* IDCC 3551.

**Figure 2 microorganisms-10-02218-f002:**
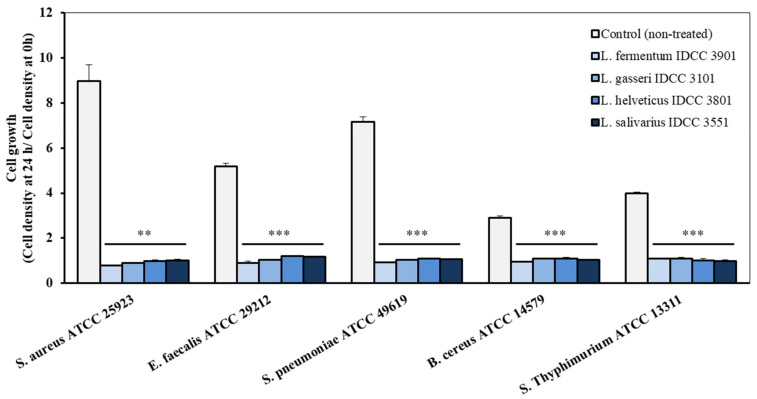
Antimicrobial effect of the cell-free supernatant of the four *Lactobacillus* strains on pathogenic microorganisms. (**, *p* < 0.01; ***, *p* < 0.001 vs. control group).

**Table 1 microorganisms-10-02218-t001:** Probiotic strains used in this study.

Strain	Origin	ATCC Number
*Lactobacillus fermentum* IDCC 3901	Homemade cheese	BAA-2842
*Lactobacillus gasseri* IDCC 3101	Breast milk	BAA-2841
*Lactobacillus helveticus* IDCC 3801	Breast-fed infant feces	BAA-2840
*Lactobacillus salivarius* IDCC 3551	Healthy child saliva	BAA-2835

**Table 2 microorganisms-10-02218-t002:** Pathogenic bacteria used in this study.

Strain	Culture Condition
*Staphylococcus aureus* ATCC 25923	TSB medium (BD Difco), 37 °C, 24 h, aerobic
*Enterococcus faecalis* ATCC 29212	BHI medium (BD Difco), 37 °C, 24 h, aerobic
*Streptococcus pneumoniae* ATCC 49619	BHI medium, 37°C, 24 h, aerobic
*Bacillus cereus* ATCC 14579	Nutrient medium (BD Difco), 30 °C, 24 h, aerobic
*Salmonella* Typhimurium ATCC 13311	Nutrient medium, 37°C, 24 h, aerobic

**Table 3 microorganisms-10-02218-t003:** Genomic information of four *Lactobacillus* species.

	*Lactobacillus fermentum* IDCC 3901	*Lactobacillus gasseri* IDCC 3101	*Lactobacillus helveticus* IDCC 3801	*Lactobacillus salivarius* IDCC 3551
Identification	*Limosilactobacillus fermentum*	*Lactobacillus gasseri*	*Lactobacillus helveticus*	*Lactobacillus salivarius*
Genome size (bp)	2,094,262	2,167,634	2,204,078	1,507,589
GC contents (%)	51.81	34.72	36.81	32.89
CDS	2051	2185	2085	1309
ANI value (%)	99.35	99.91	98.16	99.98

**Table 4 microorganisms-10-02218-t004:** Minimal inhibitory concentrations and antibiotic resistance genes of four *Lactobacillus* species.

Strain	Items	Antibiotics ^1^								
		AMP	VAN	GEN	KAN	STR	ERY	CLI	TET	CHL
*L. fermentum* IDCC 3901	Cut-off value (µg/mL) ^2^	2	n.r. ^3^	16	64	64	1	4	8	4
	Observed MIC	<0.125	512>	64	256	128	0.25	<0.125	2	4
	Assessment	S ^4^	n.r.	R ^5^	R	R	S	S	S	S
	Antibiotic resistance gene	n.d. ^6^	n.d.	n.d.	n.d.	n.d.	n.d.	n.d.	n.d.	n.d.
*L. gasseri* IDCC 3101	Cut-off value (µg/mL) ^2^	1	2	16	64	16	1	4	4	4
	Observed MIC	0.5	2	256	512>	256	<0.125	4	1	4
	Assessment	S	S	R	R	R	S	S	S	S
	Antibiotic resistance gene	n.d.	n.d.	n.d.	n.d.	n.d.	n.d.	n.d.	n.d.	n.d.
*L. helveticus* IDCC 3801	Cut-off value (µg/mL) ^2^	2	2	16	16	16	1	4	4	4
	Observed MIC	<0.125	1	64	128	8	<0.125	1	0.5	2
	Assessment	S	S	R	R	S	S	S	S	S
	Antibiotic resistance gene	n.d.	n.d.	n.d.	n.d.	n.d.	n.d.	n.d.	n.d.	n.d.
*L. salivarius* IDCC 3551	Cut-off value (µg/mL) ^2^	4	n.r.	16	64	64	1	4	8	4
	Observed MIC	0.25-0.5	512>	32	512>	256	0.25	0.25	1	2
	Assessment	S	n.r.	R	R	R	S	S	S	S
	Antibiotic resistance gene	n.d.	n.d.	n.d.	n.d.	n.d.	n.d.	n.d.	n.d.	n.d.

^1^ AMP—ampicillin; VAN—vancomycin; GEN—gentamicin; KAN—kanamycin; STR—streptomycin; ERY—erythromycin; CLI—clindamycin; TET—tetracycline; CHL—chloramphenicol; ^2^ European Food Safety Authority (EFSA), 2018. EFSA Journal 16(3): 5206. ^3^ n.r. not required; ^4^ S susceptible; ^5^ R resistant; ^6^ n.d. not detected.

**Table 5 microorganisms-10-02218-t005:** Enzymatic activities of four *Lactobacillus* species.

	*L. fermentum* IDCC 3901	*L. gasseri* IDCC 3101	*L. helveticus* IDCC 3801	*L. salivarius* IDCC 3551
Alkaline phosphatase	−	−	−	−
Esterase	+	+	+	−
Esterase lipase	−	−	−	−
Lipase	−	−	−	−
Leucine arylamidase	+	+	+	+
Valine arylamidase	+	−	−	+
Cystine arylamidase	−	−	+	+
Trypsin	−	−	−	−
α-Chymotrypsin	−	−	−	−
Acid phosphatase	+	+	+	+
Naphthol-AS-BI-phosphohydrolase	−	+	+	+
α-Galactosidase	+	+	−	+
β-Galactosidase	+	+	+	+
β-Glucuronidase	−	−	−	−
α-Glucosidase	+	−	−	−
β-Glucosidase	−	+	+	−
N-Acetyl-β-glucosaminidase	−	+	−	−
α-Mannosidase	−	−	−	−
α-Fucosidase	−	−	−	−

− negative; + positive.

**Table 6 microorganisms-10-02218-t006:** API 50 CH test result of the four *Lactobacillus* strains tested in this study.

	*L. fermentum* IDCC 3901	*L. gasseri* IDCC 3101	*L. helveticus* IDCC 3801	*L. salivarius* IDCC 3551
Glycerol	−	−	−	−
Erythritol	−	−	−	−
D-Arabinose	−	−	−	−
L-Arabinose	+	−	−	−
Ribose	+	−	−	−
D-Xylose	−	−	−	−
L-Xilose	−	−	−	−
Adonitol	−	−	−	−
β-Methyl-xylose	−	−	−	−
Galactose	+	+	+	+
D-Glucose	+	+	+	+
D-Fructose	+ ^w^	+	+	+
D-Mannose	−	+	+	+
L-Sorbose	−	−	−	−
Rhamnose	−	−	−	−
Dulcitol	−	−	−	−
Inositiol	−	−	−	−
Mannitol	−	−	−	+
Sorbitol	−	−	−	+
α-Methyl-D-mannoside	−	−	−	−
α-Methyl-D-glucoside	−	−	−	−
N-Acethyl-Glucosamine	−	+	+	+
Amygdaline	−	+ ^w^	+	−
Arbutine	−	+	+	+
Esculine	+	+	+	+
Salicine	−	+	+	+
Cellobiose	−	+	+	−
Maltose	+	+	+	+
Lactose	+	+	+	+
Melibiose	+	−	−	+
Sucrose	+	+	+	+
Trehalose	−	+	+	+
Inuline	−	−	−	−
Melizitose	−	−	−	−
D-Raffinose	+	−	−	+
Amidon	−	+ ^w^	+ ^w^	−
Glycogene	−	−	−	−
Xylitol	−	−	−	−
Gentibiose	−	+	+	−
D-Turanose	−	+	−	−
D-Lyxose	−	−	−	−
D-Tagatose	−	+	−	−
D-Fucose	−	−	−	−
L-Fucose	−	−	−	−
D-Arabitol	−	−	−	−
L-Arabitol	−	−	−	+
Gluconate	+ ^w^	−	−	−
2-keto-gluconate	−	−	−	−
5-keto-gluconate	+ ^w^	−	−	−

− negative; + positive; + ^w^ weak positive.

**Table 7 microorganisms-10-02218-t007:** Body weight changes of rats administered the four *Lactobacillus* species.

Strains	Group	Dose (g/kg BW ^1^)	Day after Administration				
			0	1	3	7	14
*L. fermentum* 2232IDCC 3901	9 weeks old	300	216.0 ± 2.9	233.0 ± 11.5	246.4 ± 12.5	257.4 ± 20.3	264.1 ± 13.3
		2000	209.9 ± 8.6	231.4 ± 12.6	235.3 ± 15.0	243.1 ± 16.9	253.3 ± 12.4
	10 weeks old	300	226.4 ± 3.0	248.5 ± 9.6	251.9 ± 6.5	260.3 ± 5.3	268.7 ± 4.4
		2000	217.6 ± 10.7	240.6 ± 10.0	246.7 ± 9.1	252.9 ± 13.3	260.2 ± 18.2
*L. gasseri* 2232IDCC 3101	9 weeks old	300	216.6 ± 14.5	231.9 ± 11.6	248.6 ± 10.6	259.2 ± 11.1	270.9 ± 16.0
		2000	202.3 ± 3.2	224.7 ± 9.9	230.8 ± 8.0	236.8 ± 11.4	251.6 ± 4.9
	10 weeks old	300	225.0 ± 3.7	241.5 ± 9.6	257.9 ± 8.5	259.6 ± 2.9	270.8 ± 6.3
		2000	222.1 ± 4.1	251.6 ± 7.0	257.6 ± 6.0	265.0 ± 4.3	269.0 ± 6.7
*L. helveticus* 2302IDCC 3801	9 weeks old	300	222.6 ± 8.6	247.2 ± 8.3	255.8 ± 4.8	270.4 ± 10.9	279.1 ± 14.0
		2000	217.1 ± 5.1	231.7 ± 6.1	237.8 ± 1.9	243.8 ± 8.2	254.5 ± 7.5
	10 weeks old	300	245.3 ± 5.3	268.5 ± 7.7	273.5 ± 5.2	277.6 ± 10.5	294.3 ± 11.4
		2000	222.2 ± 4.2	242.7 ± 10.5	254.2 ± 8.0	264.5 ± 4.1	271.2 ± 9.5
*L. salivarius* IDCC 3551	9 weeks old	300	207.6 ± 3.8	231.4 ± 5.4	236.4 ± 1.4	241.6 ± 2.1	264.4 ± 2.8
		2000	209.1 ± 7.2	234.7 ± 9.3	237.7 ± 9.3	250.8 ± 13.8	259.2 ± 15.7
	10 weeks old	300	222.8 ± 5.4	245.5 ± 6.5	251.6 ± 1.3	262.3 ± 2.0	269.6 ± 5.5
		2000	218.5 ± 5.1	244.3 ± 5.6	248.5 ± 0.8	258.2 ± 0.8	266.1 ± 4.1

^1^ BW body weight.

## Data Availability

Not applicable.

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
