# Peer review of "Safety Evaluation by Phenotypic and Genomic Characterization of Four *Lactobacilli* Strains with Probiotic Properties"

_microorganisms, 2022, doi:10.3390/microorganisms10112218_

Round 1
Reviewer 1 Report
My comments are as below:
The manuscript seems unfinished. The abstract must contain information about the authors' significant results in terms of information about significant discoveries. The opening must provide a coherent hypothesis and substantially develop the second paragraph. There is unnecessary repeating of material throughout the text. Check figure ligands; they are written clumsily. The discussion should provide additional information and references pertaining to pertinent and related literature. Reorganize and thoroughly revise the conclusion. There should be a separate discussion.
Reviewer 2 Report
The manuscript is devoted to the study of four lactobacilli as potential agents for probiotic preparations.
The article corresponds to the profile of the journal Microorganisms. In general, the authors presented in the article the results corresponding to the task of the study. There is a small note - the authors are requested to describe in more detail the preparation of culture for testing.
Line 93. It's written "Each Lactobacillus species was grown overnight" What was the medium? OD?
Lines 98-100 "The culture of each strain was harvested and used to prepare a culture solution. The culture solution was inoculated into tubes containing the substrates and incubated at 37 °C for 4 h." Questions: 1. Was harvected from what medium? 2. How culture sol-n was prepared? How much of culture solution was inoculated?
Lines 106-108 "The culture of each strain was harvested and used to prepare culture solutions. The culture solutions were inoculated into tubes containing different carbohydrates..." Questions are the same: 1. Was harvected from what medium? 2. How culture sol-n was prepared? How much of culture solution was inoculated?
The title of figure one is proposed to be changed from "hemolytic activity of lactobacillus" to "test for hemolytic activity", since none of the tested strains showed such activity.
Reviewer 3 Report
This study provides a way to assess the potential risks of four different Lactobacillus species and validate the safety of all four strains as probiotics for human consumption. Indeed, for each probiotic strain, it is necessary to check for their safety. Indeed, the study of the genome of probiotic strains allows us to assess the presence of pathogenicity and antibiotic resistance genes. In addition, the authors propose an original way of assessing toxicity in vitro and in vivo.
Title of the article "Phenotypic and genomic characterization of four Lactobacilli strains with probiotic properties" involves the study of the probiotic properties of lactobacilli. However, not all lactobacilli are ideal for creating probiotics. The authors do not describe the beneficial properties of the studied strains. For example, their antagonistic activity against bacteria, fungi and viruses, which could be associated, for example, with the presence in their genome of genes encoding bacteriocins, other antimicrobial factors. Phenotypic effects confirming the presence of antimicrobial activity in these strains of lactobacilli, primarily in relation to pathogens, have not been described. Their effect on the immune system is also not specified. It would be interesting to know other potential effects of these probiotic strains on the nervous system and metabolism.
It is not clear whether such studies have been conducted before. The authors do not indicate the source of isolation of these strains and why these strains were chosen for the study. Whether these lactobacilli strains were used for medical purposes?
Phenotypic effects confirming the presence of antimicrobial activity in these strains of lactobacilli, primarily in relation to pathogens, have not been described. Their effect on the immune system is also not specified. It would be interesting to know other potential effects of these probiotic strains on the nervous system and metabolism.
The title of the article is unsuccessful because the study primarily concerns the methodology for studying the safety of strains, in particular, the lack of transmission of antibiotic resistance genes and the possibility of adverse effects on the body due to toxicity (lack of hemolytic activity and genes encoding toxins), colonization of the body due to high adhesiveness.
It would be better to emphasize in the title of paper that the safety of probiotic strains was mainly investigated.
The paper does not indicate which pathogenicity genes were analyzed. They should be specified in the methods section or the description of the results.
It should be noted that it is quite difficult to raise the question of the presence of pathogenicity genes in lactobacilli. It is known that despite the fact that lactobacilli are more often considered as beneficial bacteria. However, even probiotic strains used in medicine are not always safe. In some cases, they can cause serious complications and side effects. Currently, the pathogenic potential of lactobacilli remains debatable and relatively poorly understood. It is not entirely clear which publications on pathogenicity genes and phenotypic manifestations of pathogenicity the authors focus on. They are not specified in the article.
To study the safety of probiotic strains, experiments are required to identify acute and subacute toxicity of strains, immunotoxicity and embrotoxicity, which are currently mandatory for preclinical studies of all probiotic strains. The authors could at least mention this.
At the same time, the proposed methods and the data obtained on the study of acute oral toxicity and the study of the hemolytic activity of lactobacilli are useful and can be recommended for the study of the safety of probiotic strains. The study of the safety of probiotic strains is usually preceded by the establishment of their beneficial properties for use in medicine. This is exactly what should have been written in the introduction to this manuscript and the description of the materials in the "Materials and methods" section.
The absolute advantage of this work is the phenotypic study of the enzymatic activity of the analyzed strains of lactobacilli and the corresponding genes. In this case, significant differences between them are revealed. However, conclusions on their possible specific use in the presence of metabolic disorders in the host body are not given.
The source of strain isolation is unclear. What is the reason for their choice . The introduction emphasizes that strains isolated from different sources and with different strain characteristics can be used for targeted therapy. In conclusion, it would be interesting to compare the analyzed strains, and not only to indicate that their safety has been proven. Using previously obtained data on the probiotic potential of these bacteria and the features of their metabolic activity identified by the authors, it would be advantageous to indicate the directions of further research of the analyzed lactobacilli.
The safety of these bacteria could also be confirmed later using other methods as part of a preclinical study.

Round 2
Reviewer 1 Report
Manuscript can be accepted for publication.
Author Response
We thank the reviewer for the encouragement and comments made about this manuscript.
Reviewer 3 Report
The authors have significantly changed the text of the article. It has become more logical and more understandable for considering the beneficial properties of probiotic strains. However, I did not find a list of lactobacillus pathogenicity factors that were analyzed. It is also not entirely clear whether these strains were used for medical purposes as probiotics.
